

# Favour the best in case of emergency cricothyroidotomy–a randomized cross-over trial on manikin focused training and simulation of common devices

Nicole Didion[1], Fabian Pohlmann[1], Nina Pirlich[1], Eva Wittenmeier[1], Christoph Jänig[2], Daniel Wollschläger[3] and Eva-Verena Griemert[1]

[1] Department of Anaesthesiology, Johannes-Gutenberg Universität Mainz, Mainz, Rhineland-Palatinate, Germany
[2] Department of Anaesthesiology and Critical Care Medicine, Central hospital of the German armed forces, Koblenz, Rhineland-Palatinate, Germany
[3] Institute of Medical Biostatistics, Epidemiology and Informatics, University Medical Centre of the Johannes Gutenberg-University Mainz, Mainz, Rhineland-Palatinate, Germany

Corresponding author
Nicole Didion, didion@uni-mainz.de

## ABSTRACT

**Background:** Performing an emergency cricothyroidotomy (EC) is extremely challenging, the devices used should be easy to handle and the selected technique reliable. However, there is still an ongoing debate concerning the most superior technique.

**Methods:** Three different techniques were compared using a standardized, simulated scenario regarding handling, performing, training and decision making: The scalpel-bougie technique (SBT), the surgical anatomical preparation technique (SAPT) and the Seldinger technique (ST). First, anaesthesia residents and trainees, paramedics and medical students (each group $n = 50$) performed a cricothyroidotomy randomly assigned with each of the three devices on a simulator manikin. The time needed for successful cricothyroidotomy was the primary endpoint. Secondary endpoints included first-attempt success rate, number of attempts and user-satisfaction. The second part of the study investigated the impact of prior hands-on training on both material selection for EC and on time to decision-making in a simulated "cannot intubate cannot ventilate" situation.

**Results:** The simulated scenario revealed that SBT and SAPT were significantly faster than percutaneous EC with ST ($p < 0.0001$). Success rate was 100% for the first attempt with SBT and SAPT. Significant differences were found with regard to user-satisfaction between individual techniques ($p < 0.0001$). In terms of user-friendliness, SBT was predominantly assessed as easy (87%). Prior training had a large impact regarding choice of devises ($p < 0.05$), and time to decision making ($p = 0.05$; 180 s *vs.* 233 s).

**Conclusion:** This study supports the use of a surgical technique for EC and also a regular training to create familiarity with the materials and the process itself.The trial was registered before study start on 11.11.2018 at **ClinicalTrials.gov** (NCT: 2018-13819) with Nicole Didion as the principal investigator.

# INTRODUCTION

Cricothyroidotomy is a rescue procedure in a "cannot intubate cannot ventilate" (CICV) situation when other methods of nonsurgical airway management have failed. In this critical situation, the only way to protect the patient from severe harm is an immediate emergency cricothyroidotomy (EC). Although airway management is the core competency of anaesthesiologists, a CICV-situation is an extremely stressful scenario because very few clinicians have ever performed an EC (*Baker, Feinleib & O'Sullivan, 2016*). Due to the implementation of the laryngeal mask and video laryngoscopy, the need to perform EC diminished (*Frerk et al., 2015*; *Greenland et al., 2011*). In addition to practical experience, the chosen technique for EC also influences the success rate. Accordingly, the national audit project 4 (NAP 4) on airway management practice in the United Kingdom stated that EC is associated with a significant failure rate and the success rate correlates with the chosen EC technique (*Cook, Woodall & Frerk, 2016*). The various techniques differ between surgical, scalpel-based, percutaneous or needle-based approaches (*Hamaekers & Henderson, 2011*). The open surgical technique uses either the scalpel-bougie technique (SBT), the surgical anatomical preparation technique (SAPT) or a modification of these techniques. Percutaneous puncture of the cricothyroid membrane includes the Seldinger technique (ST), the wide-bore cannula-over-trocar technique (*e.g.*, Quicktrach® VBM, Sulz, Germany) or the narrow-bore cannula-over-needle technique. A third important factor affecting the success rate of EC is the cognitive process of decision-making (*Riem et al., 2012*). A well-planned approach is one of the most important human factors required for crisis management.

This study examines all three aspects that may influence the outcome of an EC. First, the issue of appropriate technique was addressed. To date, recommendations regarding the appropriate technique for performing EC have been very inconsistent (*Apfelbaum et al., 2022*; *Law et al., 2013*). The Difficult Airway Society (DAS) guidelines published in 2015 recommend the SBT (*Frerk et al., 2015*). In contrast, the German S1 guideline on airway management does not recommend a specific technique due to lack of evidence (*Piepho et al., 2015*). In our study, three established cricothyroidotomy techniques were compared in a simulated manikin situation. Parallel we also investigated the impact of a prior EC-training during the study as a part of a simulation scenario. We compared participants who had already performed EC using the described devises to those without prior training with regard to the time for the decision-making process.

# MATERIALS AND METHODS

After approval of the study by the local Research Ethics Committee of the state medical association (Mainz, Rhineland-Palatinate, Germany–Application number: 2018-13819) written informed consent was obtained from each participant. The participants were on the one hand medical students or employees at the University Medical Centre of the

Johannes Gutenberg-University Mainz, Germany. Paramedics on the other hand, were from the local ground-and air-based emergency medical service.

We recruited anaesthesiologists at all levels of training, medical students and paramedics. The following data were collected *via* a questionnaire: profession, professional experience, known cricothyroidotomy techniques, preferred technique for EC, previous experience with EC in training situations, previous experience and chosen technique in a real world EC procedure.

The following three techniques were compared in a randomized cross-over study design:

SBT: The ScalpelCric® (VBM, Sulz am Neckar, Germany) is a surgical set for cricothyroidotomy developed to match the latest recommendations of the DAS 2015 guidelines and is based on the simple description of "stab, twist, bougie, tube". The set contains a scalpel Number 10, a 14 French 40 cm length bougie with an option for oxygen delivery through the bougie, a cuffed endotracheal tube (ID 6.0 mm), an extension tubing, a 10 ml syringe and a neck tape.

SAPT: After identification of anatomical landmarks, a vertical incision was made through skin and subcutaneous tissue using a no. 11 scalpel blade (Feather® Safety Razor Co.). The cricothyroid membrane was incised horizontally and the resulting hole was enlarged and held open with a 14 cm speculum (Medicon eG, Tuttlingen, Germany). A cuffed endotracheal tracheal tube (ID 5.0 mm, OD 6.7 mm, Rüschelit® Super Safety Clear, Teleflex Medical GmbH, Fellbach, Germany) was inserted into the trachea and the inflation of the cuff completed the procedure.

ST: The Melker Emergency Cricothyroidotomy Catheter Set (Cook® Medical Inc., Bloomington, Indiana, USA) contains two kinds of introducer needles, a syringe, a scalpel Number 15, a guide wire, a curved dilatator, a tracheostomy tape and an airway catheter (ID 5.0 mm, OD 7.2 mm, length 9.0 cm).

Part A: First, anaesthesia trainees, anaesthesia residents, paramedics, and medical students (each group $n = 50$) received a written and video-based introduction of each technique and device.

All 200 participants performed a cricothyroidotomy using the Crico-Trainer "Adelaide"® (VBM Medizintechnik GmbH, Sulz am Neckar, Germany) with each of the three options (SBT, SAPT, ST) randomly assigned. The correct performance of the cricothyroidotomy and position of the tube was supervised by two instructors and verified by using the Ambu® aScope™ (Ambu GmbH, Bad Nauheim, Germany). Primary outcome was the time required to successful cricothyroidotomy. This was defined as the time from touching one of the devices until tracheal placement of tube. A procedure time exceeding 360 s was defined as a failed attempt. Secondary endpoints were first-attempt success rate, number of attempts, and user-satisfaction. User-satisfaction was assessed using a four-point-scale Likert scale: 1 = very easy, 2 = easy, 3 = difficult, 4 = very difficult (*Jebb, Ng & Tay, 2021*). In the pre-study questionnaire, all participants had to indicate their preferred technique for EC, to examine whether training had an influence on the previously preferred technique.

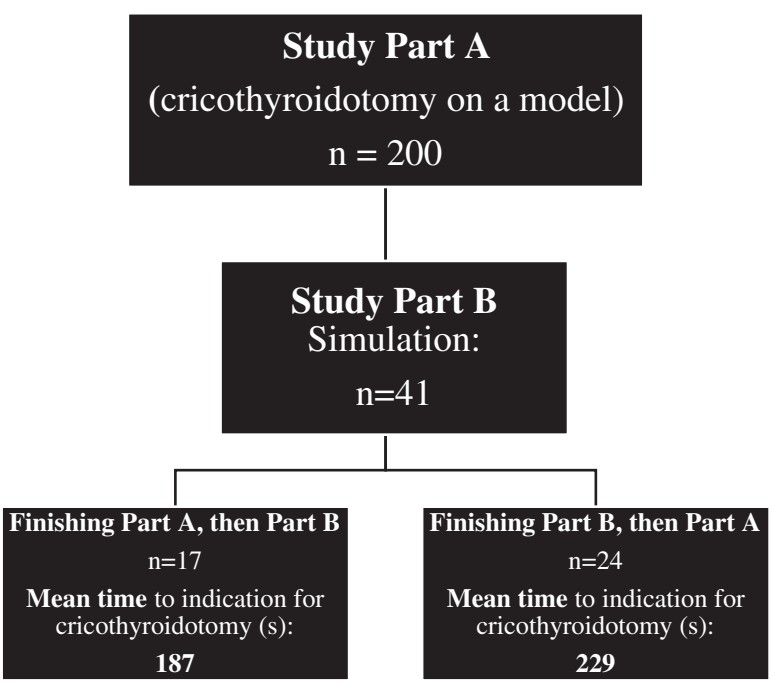

**Figure 1 Description of the study procedure-part A and part B (Scalpel-bougie technique (SBT), surgical anatomical preparation technique (SAPT) and Seldinger technique (ST)), including mean time (s) to indication for cricothyroidotomy in Part B.**

Part B: The simulation scenario was a CVCI situation due to airway obstruction caused by a wasp sting into the soft palate using the Megacode Kelly advanced Simpad training manikin® (Laerdal Medical AS, Stavanger; Norway). A total of 41 anaesthesia trainees or residents participated in Part A and B (Fig. 1). A total of 17 participants performed the described CVCI scenario on the manikin before cricothyroidotomy training. A total of 24 participated first in study part A, they were therefore familiar with the techniques before performing the CVCI simulation scenario. The endpoints of part B were the impact of a refreshed training on the preferred technique in the simulation scenario and the time to reach the decision to perform an EC. A detailed description of the two parts is illustrated in Fig. 1.

The CONSORT flow diagram, recommended for randomized controlled trials, can be found in Fig. 2.

# RESULTS

Statistical analyses were performed using SPSS (Chicago, IL, USA) and the R environment for statistical computing (version 4.1.2). Descriptive results are presented using absolute and relative frequencies for categorical variables, as well as means, medians and standard deviations for continuous variables.

In part A, median time to successful EC was calculated using the Kaplan-Meier estimator including the 95% confidence interval (CI) after right-censoring attempts which took longer than 360 s. The association of time to successful EC with each of the three above described techniques was analysed using multivariable Cox regression with a robust
## CONSORT 2010 Flow Diagram

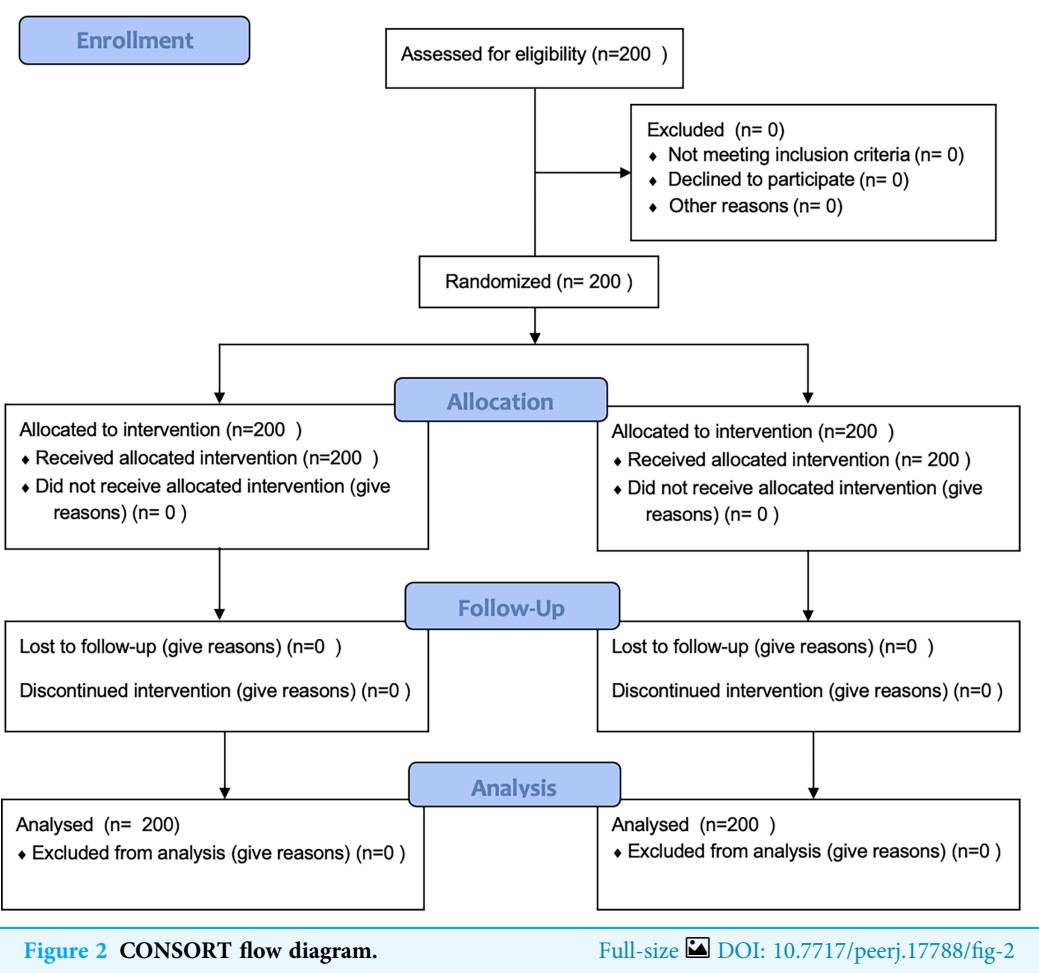

**Figure 2 CONSORT flow diagram.**

sandwich estimator for the variance to account for dependencies due to the repeated-measures design. Covariates were the qualification of the study participant (anaesthesiology resident, anaesthesiology trainee, medical student, paramedic), and the technique (SBT, SAPT, ST). Tukey contrasts with $p$-value adjustment were used for pairwise comparisons between groups. User satisfaction was compared between techniques using a multivariable mixed ordinal logistic regression model with per-person random intercepts. Fixed effects were the qualification of the study participant and the technique. The likelihood ratio test for dropping technique from the model is reported.

In part B, time to reach the decision to perform EC was analysed using multivariable Cox regression. Covariates were participant qualification, having had pre-simulation training, and the interaction term of both variables. We report adjusted hazard ratios and 95% Wald-type confidence intervals. Associations between categorical pre-simulation and post-simulation variables were assessed using Cochran's Q test.

For all tests, a $p$-value of $<0.05$ was considered significant.

**Table 1 Description of the participants, their preferred technique before study start and their experience (years–mean value) of emergency cricothyroidotomy (EC).**

| | Professional experience (years) | Emergency medicine | Preferred technique before study start | Experience with EC (any) | | |
| --- | --- | --- | --- | --- | --- | --- |
| | | | | Simulator training <= 1 year | Simulator training >1 year | During clinical CICV situation |
| Anaesthesia residents (n = 50 (%)) | 10 | 50 (100%) | None: 1 (2%)<br>SBT: 2 (4%)<br>SAPT: 39 (78%)<br>ST: 6 (12%)<br>Other: 2 (4%) | 13 (26%) | 36 (72%) | 14 (28%) |
| Anaesthesia trainees (n = 50 (%)) | 4 | 29 (58%) | None: 2 (4%)<br>SBT: 4 (8%)<br>SAPT: 33 (66%)<br>ST: 10 (20%)<br>Other: 1 (2%) | 17 (34%) | 5 (10%) | 6 (12%) |
| Paramedics (n = 50 (%)) | 7.5 | 50 (100%) | None: 8 (16%)<br>SBT: 1 (2%)<br>SAPT: 22 (44%)<br>ST: 5 (10%)<br>Other: 14 (28%) | 7 (14%) | 3 (6%) | 3 (6%) |
| Students (n = 50 (%)) | None | 21 (42%) | None: 30 (60%)<br>SBT: 0 (0%)<br>SAPT: 9 (18%)<br>ST: 3 (6%)<br>Other: 8 (16%) | 4 (8%) | 2 (4%) | None |

**Part A-participant characteristics:** Characteristics of the 200 participants including their professional experience are listed in Table 1. Prior to the study start, 15 anaesthesia residents, nine anaesthesia trainees, five paramedics and two students were familiar with the SBT from the literature (Table 2).

A total of 600 EC procedures were performed by 200 participants, of whom 599 were successful on the first attempt. One participant required a second attempt for a successful cricothyroidotomy with the ST. Time for successful cricothyroidotomy with the SBT was faster (median 39.0 s, 95% CI [37.2–42.3 s]) by 193 of 200 participants compared to the ST (median 70.4, 95% CI [65.8–77.5 s]), and for 148/200 slower compared to the SAPT (median 32.0 s, 95% CI [29.8–36.0 s]) (Table 3). The largest differences were observed in the paramedic and student group. In multivariable Cox regression, SBT was associated with significantly faster EC compared to ST ($p < 0.0001$), and with significantly slower EC compared to SAPT ($p = 0.0001$), which in turn was also significantly faster than SBT ($p < 0.0001$).

**Table 2** Description of the preexisting practical experience regarding the described techniques (Scalpel bougie technique (SBT); surgical anatomical preparation technique (SAPT); Seldinger technique (ST)) separated according to participant groups.

|  | Total n = 200 | Anaesthesia residents n = 50 | Anaesthesia trainees n = 50 | Paramedics n = 50 | Students n = 50 |
|---|---|---|---|---|---|
| SBT | 15 (7,5%) | 10 (20%) | 3 (6%) | 2 (4%) | 0 (0%) |
| SAPT | 84 (42%) | 42 (84%) | 34 (68%) | 4 (8%) | 4 (8%) |
| ST | 89 (44,5%) | 38 (76%) | 30 (60%) | 16 (32%) | 5 (10%) |

**Table 3** Overall results (in s) regarding scalpel-bougie technique (SBT), surgical anatomical preparation technique (SAPT) and Seldinger technique (ST) with mean time to successful airway access, standard deviation (SD).

|  | SBT | | SAPT | | ST | |
|---|---|---|---|---|---|---|
|  | Time (s) | SD | Time (s) | SD | Time (s) | SD |
| Overall results | 43.6 | 19.2 | 35.8 | 16.4 | 76.8 | 32.6 |
| Subgroups | | | | | | |
| Anaesthesia residents | 35.4 | 12.0 | 24.9 | 11.2 | 53.7 | 15.7 |
| Anaesthesia trainees | 36.0 | 10.9 | 32.0 | 13.2 | 57.7 | 18.3 |
| Paramedics | 49.5 | 18.0 | 41.2 | 17.3 | 96.8 | 32.5 |
| Students | 53.4 | 25.6 | 44.9 | 15.6 | 98.9 | 29.3 |

Across all professional groups, there was a positive effect of previous technique-specific EC training on duration in part A, most notably with the ST (ST: $p < 0.0001$, SAPT: $p < 0.0001$, SBT: $p = 0.9$) (Table 4). A statistically significant difference was also observed with respect to the occurrence of any kind of EC-training, regardless of technique, within ≤1 year prior to the start of the study (44.5 s *vs.* 54.1 s, $p = 0.004$).

There were also statistically significant differences between techniques with respect to user satisfaction ($p < 0.0001$). While ratings for SBT (mean = 1.8, SD = 0.7) and SAPT (mean = 1.9, SD = 0.8) were similar, those for ST were inferior (mean = 2.5, SD = 0.9). Occupational groups had no statistically significant difference in reported user-satisfaction ($p = 0.37$). Detailed data divided by individual technique and professional groups are shown in Table 5.

In the pre-test questionnaire, primarily students and paramedics were not able to indicate a preferred EC technique due to lack of experience with EC. Twenty-five (12.5%) of the participants preferred a device (Quicktrach®) that was not used in this study. Seventy (35.0%) of the participants were influenced by the training and changed their preference in the study.

**Part B:** Of the 24 anaesthesiologists trained prior to the simulation scenario, 23 (95.8%) completed the scenario, and all of the 17 anaesthesiologists without prior training successfully completed the scenario. On average, prior training reduced the time required for decision making and establishing the EC ($p = 0.04$; Fig. 1). Anaesthesia residents reached the decision to perform EC significantly faster than anaesthesia trainees ($p = 0.01$).

**Table 4  Mean time to successful airway access (in s) with standard deviation (SD) and *p*-value for comparison stratified by technique specific emergency-cricothyroidotomy training (EC-training) conducted ≤12/>12 month before study start.**

|  | SBT | | | SAPT | | | ST | | |
|---|---|---|---|---|---|---|---|---|---|
|  | Time (s) | SD | *p*-value | Time (s) | SD | *p*-value | Time (s) | SD | *p*-value |
| EC-training ≤12 months | 42.3 | 19.1 | 0.9 | 26.7 | 12.7 | <0.0001 | 55.8 | 33.2 | <0.0001 |
| EC-training >12 months | 43.6 | 19.3 |  | 39.0 | 16.4 |  | 82.4 | 33.2 |  |

**Table 5  User comfort *via* likert-scale (Mean value and *p*-value for comparison) divided by individual technique and professional groups.**

|  | User comfort/Likert-scale (1–4) | | | |
|---|---|---|---|---|
|  | SBT | SAPT | ST | *p* |
| Anaesthesia residents (*n* = 50) | 2 | 1.92 | 2.26 | <0.001 |
| Anaesthesia trainees (*n* = 50) | 1.82 | 1.96 | 2.12 | <0.001 |
| Paramedics (*n* = 50) | 1.8 | 1.76 | 2.86 | <0.001 |
| Students (*n* = 50) | 1.6 | 2.02 | 2.78 | <0.001 |

There was no statistically significant evidence for a differential training effect between anaesthesia residents and trainees ($p = 0.44$).

# DISCUSSION

There are different techniques to perform an EC. The present study comprised 200 participants to compare two surgical methods (SBT and SAPT) and one puncture method (ST) with respect to various aspects of feasibility using a standard model. In addition, the study investigated the impact of professional qualificational level of training on the performance of EC. The results show that on a validated cricothyroidotomy trainer the surgical approach, *i.e.*, SAPT and the SBT, was significantly faster than the needle-based ST, regardless of medical occupation and professional experience ($p < 0.05$; Table 2). If one compares only the two surgical approaches, SAPT was superior to SBT. Training had a significant positive impact on performance with all techniques and participants.

Several studies focused on the time needed for cricothyroidotomy, comparing surgical techniques with the percutaneous Seldinger technique (*Andresen, Kramer-Johansen & Kristiansen, 2019*; *Chrisman et al., 2016*; *Dimitriadis & Paoloni, 2008*; *Eisenburger et al., 2000*; *Schaumann et al., 2005*). *Mariappa et al. (2009)* found no significant difference in the completion time comparing the surgical technique with the Seldinger technique and the Portex Cricothyroidotomy Kit®. Regarding this study, it should be noted that, in contrast to ours, it consisted solely of four participating intensivists from various backgrounds and with differing levels of experience. However, comparing the Seldinger technique with a standard scalpel-based approach performed by 20 almost unexperienced emergency physicians found a significantly faster completion with the Seldinger technique (*Schaumann et al., 2005*). This might be caused by different tube sizes used per approach. A majority of studies ascribed the surgical approach to be more rapid in contrast to the

Seldinger technique, independent of the model and the experience of participants (*Andresen, Kramer-Johansen & Kristiansen, 2019*; *Chang et al., 2018*; *Chrisman et al., 2016*; *Dimitriadis & Paoloni, 2008*; *Poole et al., 2017*; *Salah et al., 2010*; *Schober et al., 2009*; *Sulaiman, Tighe & Nelson, 2006*). This is in accordance with the results of the present study.

In the CICV setting, not only the time until successful EC is crucial, but also the success-rate of the first attempt and the total number of attempts. The literature shows inconsistent results. In a systematic review comprising 24 prospective experimental studies, no difference in the success-rates between the open surgical technique and the percutaneous puncture technique was found (*Langvad et al., 2013*). In human cadavers, 63 inexperienced health care workers were successful in 94% of the cases with the surgical technique, whereas the success rate for the wire-guided ST technique was only 71% (*Schober et al., 2009*). In contrast, experienced airway-managers working in an ICU had a success-rate of only 55% using the surgical method, while 100% were successful using the ST (*Mariappa et al., 2009*). Experienced airway-managers are most likely familiar with percutaneous dilated tracheostomy and are therefore probably more comfortable with the needle-based approach. In the present study, only one out of 200 participants needed a second attempt with the ST. All other completed the EC within the first attempt. The success-rates of the EC methods tested in heterogeneous studies are only comparable to a limited extent because different airway models were used with differences in material and design (*e.g.*, human cadavers, animal models, and a multitude of airway models). In addition, diverse definitions of EC start and end points, heterogeneous distribution of participants, and small sample sizes make it difficult to compare study results. This should be considered when evaluating different EC methods. To generate highly reliable results, the present study included 200 participants from four pre-defined medical groups whose performance on a total of 600 ECs was assessed using well-defined endpoints.

User satisfaction is a crucial factor depending on how comfortable the user felt with the procedure. Surprisingly, anaesthesiologists felt more confident with the open surgical approach than with ST despite the fact that anaesthesiologists in general are very familiar with the "Seldinger technique" in other settings. In the paramedic and medical student group, the majority had no prior experience with EC. They clearly preferred SBT and SAPT. This is consistent with the results of other studies in which the surgical approach was preferred (*Chang et al., 2018*; *Chrisman et al., 2016*; *Heymans et al., 2016*). However, many factors such as user experience, the complexity and error-proneness of the application influence user-satisfaction. Therefore, the available techniques should be even more intuitive and trained frequently on suitable models. Several authors highlighted the relevance of regular training to manage a CICV-situation, especially to internalize algorithms and complex procedural skills (*Boet et al., 2011*; *Chang et al., 2018*; *Hossfeld et al., 2019*; *Hubert et al., 2014*). The results of the present study showed that training shortens the time to decision to perform EC and also significantly reduced the time to completion of the EC-procedure in a simulated CICV-Situation.

The debate which procedure is the best for establishing an EC continues. The DAS Guidelines 2015 and the guidelines of the Canadian Airway Focus Group recommend a

surgical approach at first (*Frerk et al., 2015*; *Law et al., 2013*). In the present study only a minority of participants (7%) was familiar with the scalpel-bougie technique, and only one participant chose this technique for the simulation-study (part B) from the beginning. After previous training, 14% preferred the scalpel-bougie technique as their first line technique. In accordance with the DAS-recommendations we showed that the scalpel-bougie technique is easy to learn and the success-rate and time needed is comparable to the conventional surgical technique. As the incidence of real-world CICV-scenarios is rather low, they are associated with a high-complication rate up to 40%, and failure of the procedure is life-threatening (*Helm et al., 2013*). Therefore, improving the competence of airway management in terms of procedure and material familiarity as well as soft skills such as clear team-communication are just as important as the question of the optimal technique.

Practice on a low-fidelity cricothyroidotomy trainer is a limitation of the study, as this model does not represent every variation of human anatomy, complications as *e.g.*, severe bleeding and the lack of resemblance to human tissue. These are the disadvantages of the utilized model and may also be an explanation of the high success rate on the first attempt. However, in contrast to cadaver-based trainings, which is hardly feasible with such a high number of cases the method ensures standardised conditions for all participants. Furthermore, the consistently identical conditions allowed for a valid comparison of the three cricothyroidotomy techniques, which signifies an advantage of the used cricothyroidotomy trainer. Another bias that could influence the results is the variance in the OD of the tubes used in the study. Lastly, simulated scenarios can cause lower physical and psychological stress than a real emergency. Given that all participants had the same conditions, there should be no systematic influence on the study results.

## CONCLUSIONS

The study concludes that surgical cricothyroidotomy techniques, such as SBT and SAPT, demonstrate superiority in terms of the time taken to successfully perform EC compared to the percutaneous puncture technique, ST. The surgical techniques also exhibit a high success rate and are favoured by both experienced and unexperienced operators. Furthermore, the study highlights the importance of training in reducing the time taken to make a decision and complete the EC procedure in a simulated CICV-scenario. Overall, the results emphasise the crucial role of training in this rare but life-saving procedure.

## ACKNOWLEDGEMENTS

This article contains parts of the doctoral dissertation of Fabian Pohlmann and the professorial dissertation (Habilitation) of Nicole Didion.

### Funding

The authors received no funding for this work.

## Competing Interests

The authors declare that they have no competing interests.

## Author Contributions

- Nicole Didion conceived and designed the experiments, performed the experiments, analyzed the data, prepared figures and/or tables, authored or reviewed drafts of the article, and approved the final draft.
- Fabian Pohlmann performed the experiments, analyzed the data, prepared figures and/or tables, authored or reviewed drafts of the article, and approved the final draft.
- Nina Pirlich conceived and designed the experiments, performed the experiments, analyzed the data, authored or reviewed drafts of the article, and approved the final draft.
- Eva Wittenmeier analyzed the data, authored or reviewed drafts of the article, and approved the final draft.
- Christoph Jänig analyzed the data, authored or reviewed drafts of the article, and approved the final draft.
- Daniel Wollschläger analyzed the data, authored or reviewed drafts of the article, and approved the final draft.
- Eva-Verena Griemert analyzed the data, prepared figures and/or tables, authored or reviewed drafts of the article, and approved the final draft.

## Human Ethics

The following information was supplied relating to ethical approvals (*i.e.*, approving body and any reference numbers):

The Research Ethics Committee (Rhineland-Palatinate, Germany) granted Ethical approval to carry out the study within its facilities (Ethical Application Ref: 2018-13819.

## Clinical Trial Ethics

The following information was supplied relating to ethical approvals (*i.e.*, approving body and any reference numbers):

The Research Ethics Committee (Rhineland-Palatinate, Germany) granted Ethical approval to carry out the study within its facilities (Ethical Application Ref: 2018-13819.

## Data Availability

The raw data are available in the Supplemental Files.

## Clinical Trial Registration

The following information was supplied regarding Clinical Trial registration:

ClinicalTrials.gov (NCT: 2018-13819).

## Supplemental Information

Supplemental information for this article can be found online at http://dx.doi.org/10.7717/peerj.17788#supplemental-information.

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
