# Peer review of "Favour the best in case of emergency cricothyroidotomy–a randomized cross-over trial on manikin focused training and simulation of common devices"

_PeerJ, doi:10.7717/peerj.17788_

## Round 0.1 · original submission · Minor Revisions

Dear Dr. Didion,

Your manuscript entitled "Favour the best in case of emergency cricothyroidotomy – a randomized cross-over trial on manikin focused training and simulation of common devices" which you submitted to PeerJ, has been reviewed by the editor and three experts in the field.

The reviewers are in general favorable and suggest that, subject to minor revisions, your paper could be suitable for publication. Please give these points your careful attention, as the revised manuscript will undergo a second round of review by the same reviewers.

I hope that you will be prepared to make the necessary amendments and submit a revised manuscript accompanied by a statement of how you have responded to the reviewers’ comments.

Yours sincerely,
Stefano Menini

·

Basic reporting

Table 5 was not included in the review material.

The title of FIGURE 2 incorrectly showed figure 1

Figure 2 seems confusing to me.
I suggest that you divide the PART A and the PART B into two columns and indicate by arrows or balls the participants who have already had experience with simulation or the techniques

Experimental design

no comment

Validity of the findings

It was a surprise to me that:
“A total of 600 CE procedures were performed by 200 participants, of which 599 were successful on the first attempt.”
Cook et al found a high CE failure rate in NAP4 (reference n°4)
These results are probably due to the fact that VBM's Crico-Trainer -Adelaide is a model that made EC an easy maneuver.
I think you could explain the pros and cons of this model and why you chose it.

Additional comments

Congratulations on your article.
EC is a topic that must be constantly discussed.
CICO is a rare event, but with potentially catastrophic results, so we need to improve techniques and training as best we can.

Reviewer 2 ·

Basic reporting

no comment

Experimental design

no comment

Validity of the findings

no comment

Additional comments

1.in part B“The endpoints of part B were the impact of a refreshed training on the preferred technique in the simulation scenario and the time to reach the decision to perform an EC. ”how to explain this endpoint ?
2.Figure 1 does not show the different about”the time to reach the decision to perform an EC.” ,please explain that.
3.SBT: a cuffed endotracheal tube (ID 6.0 mm), an extension tubing, a 10 ml syringe and a neck tape.
SAPT: A cuffed endotracheal tracheal tube (ID 5.0 mm, OD 6.7 mm,
ST: a curved dilatator, a tracheostomy tape and an airway catheter (ID 5.0 mm, OD 7.2 mm, length 9.0 cm). three kinds of methods using different ID and OD tube may effect the time for success cricothyroidotomy

Reviewer 3 ·

Basic reporting

The question that is the basis for "emergency medicine" in Table 1 is not included in the method pre-study questionnaire (Line 134).
The contents of Line 117-180 are not listed in the measurement results in Table 3.
Although it is written in Line 203, there is no mention of "the time to reach the decision to perform an EC" in Figure 1. Since it is considered in Line 255-257, it may be a good idea to add it to Figure 1.
I think it would be better to include the data mentioned in the discussion in a table.
Reference 16 (lines 218-220) was discussed, but how about mentioning the small number of cases (four operators) as well? It may be worth emphasizing that this study is significantly smaller than the present study. Furthermore, I think the fact that the subjects were varied, with 3 specialist intensivist and 1 registrar, is consistent with the description in Line 240-242.

Experimental design

There are many examples, and I think this is an experimental system that clearly shows the changes caused by EC training.

Validity of the findings

Is it necessary to use the Kaplan-Meier estimator in Part A?

Additional comments

None in particular

---

## Round 0.2 · Minor Revisions

Dear Dr. Didion,

Thank you for your resubmission. I have now received the report from the reviewers. The revision is now acceptable for publication, but before final acceptance, I would appreciate it if you could address the remaining minor issues raised by Reviewer 2 (recheck the data in Table 2).

I hope you will make the necessary amendments and submit a revised manuscript with a statement detailing how you responded to the reviewer’s comments. Please copy and paste each reviewer’s comment above your response. Additionally, you are kindly requested to provide a complete tracked changes version of the manuscript to make verifying that the required changes have been made easier.

If you do this, I will not need to return the manuscript to the reviewers, and it can be accepted for publication.

I look forward to receiving your revision.

Stefano Menini

·

Basic reporting

The article was revised and improved by the authors' corrections

Experimental design

Design and methods are ok

Validity of the findings

Results and conclusions are ok

Additional comments

No comments

Reviewer 2 ·

Basic reporting

good

Experimental design

good

Validity of the findings

Table 2 seems confusing to me.The total number was not same with the sum of three groups, please recheck that.especially in group (SAPT); and (ST)

Additional comments

none

Reviewer 3 ·

Basic reporting

Points pointed out have been appropriately corrected. The figures and tables have also been revised to make them easier to read.

Experimental design

no comment

Validity of the findings

no comment

---

## Round 0.3 · Minor Revisions

Dear Dr. Didion, in Table 2, the percentages displayed in brackets inaccurately represent the proportion of participants relative to the total number listed in the top row of the table. This discrepancy needs to be addressed before the paper is accepted for publication.

Sincerely,
Stefano Menini

---

## Round 0.4 · accepted · Accept

Dear Dr. Didion,

Thank you for submitting the revised version of your manuscript. I have personally reviewed the revision and am satisfied with the current version. I am pleased to inform you that your manuscript is now ready for publication in PeerJ in its present form.

I want to thank all the reviewers for their efforts in improving the manuscript and the authors for their cooperation throughout the review process.

Sincerely,
Stefano Menini